# Natural Products from Tongan Marine Organisms

**DOI:** 10.3390/molecules26154534

**Published:** 2021-07-27

**Authors:** Taitusi Taufa, Ramesh Subramani, Peter T. Northcote, Robert A. Keyzers

**Affiliations:** 1School of Agriculture, Geography, Environment, Ocean and Natural Sciences (SAGEONS), Laucala Campus, The University of the South Pacific, Suva, Fiji; taitusi.taufa@usp.ac.fj; 2Ferrier Research Institute and Centre for Biodiscovery, Victoria University of Wellington, Wellington 6140, New Zealand; peter.northcote@vuw.ac.nz; 3School of Chemical and Physical Sciences, Centre of Biodiscovery, Victoria University of Wellington, Wellington 6140, New Zealand

**Keywords:** Marine Natural Products (MNPs), tropical marine organisms, Kingdom of Tonga, biological activity

## Abstract

The islands of the South Pacific Ocean have been in the limelight for natural product biodiscovery, due to their unique and pristine tropical waters and environment. The Kingdom of Tonga is an archipelago in the central Indo-Pacific Ocean, consisting of 176 islands, 36 of which are inhabited, flourishing with a rich diversity of flora and fauna. Many unique natural products with interesting bioactivities have been reported from Indo-Pacific marine sponges and other invertebrate phyla; however, there have not been any reviews published to date specifically regarding natural products from Tongan marine organisms. This review covers both known and new/novel Marine Natural Products (MNPs) and their biological activities reported from organisms collected within Tongan territorial waters up to December 2020, and includes 109 MNPs in total, the majority from the phylum Porifera. The significant biological activity of these metabolites was dominated by cytotoxicity and, by reviewing these natural products, it is apparent that the bulk of the new and interesting biologically active compounds were from organisms collected from one particular island, emphasizing the geographic variability in the chemistry between these organisms collected at different locations.

## 1. Introduction

MNPs are secondary metabolites produced by both micro- and macro- marine organisms, produced either as a result of the organism adapting to its surrounding environment or as a defense mechanism against predators to assist in its survival [1]. With more than 70% of Earth′s surface covered by oceans, which are home to phyla considered to be exclusively marine [2], it is logical that the marine environment represents an exceptional reservoir of biodiversity and, hence, biologically active natural products, many of which exhibit unique and novel chemical features.

The Kingdom of Tonga (herein referred to as Tonga) harbors an abundance of unexplored marine biomes that could be sources of new MNPs with unique biological activities. The majority of Tongan marine organisms have not been investigated for natural product biodiscovery. Until recently, access to Tonga for bioprospecting has been limited, and as such, most Pacific-located MNP research has been reported from neighboring island groups such as Fiji, Vanuatu, and the Solomon Islands. This review covers the structures and bioactivities of Tongan marine-derived NPs up to the end of 2020. Environmental influences on the production of the MNPs reviewed, and their biosyntheses, are beyond the scope of this manuscript and are not included.

## 2. The Kingdom of Tonga

Tonga is an archipelago comprising 36 inhabited and 140 uninhabited islands in the Central Indo-Pacific Ocean, situated between Fiji to the west and Samoa to the northeast. Tonga is divided into four main groups; Tongatapu, Vava‘u, Ha‘apai, and ‘Eua (Figure 1), with several other smaller groups completing the overall archipelago. The largest island, Tongatapu, on which the capital of Nuku‘alofa is located, covers 257 km^2^. Geologically, the Tongan islands are comprised of two types; most have a limestone base formed from uplifted coral formations while others consist of limestone overlaying a volcanic base. Although Tonga is not located in close proximity to the epicenter of marine biodiversity, bounded by the Philippines, the Malay Peninsula, and New Guinea [3], Tongan waters still have distinct and exceptional marine life due to their intrinsic geographical isolation and the presence of a number of major ocean currents in the region. Tonga has a total land area of 688 km^2^ with an Exclusive Economic Zone (EEZ) of 700,000 km^2^, ~1000 times more than its land area, which affords a huge diversity of marine communities in which MNP studies could be undertaken. Given the lack of industry within Tonga, most of the collection sites referred to herein can be considered as “pristine” marine ecosystems.

## 3. Tropical Marine Organisms and Biodiversity in Tongan Territorial Waters

Tropical marine ecosystems harboring a rich diversity of micro-and macro-organisms, including invertebrates, tend to produce larger quantities of structurally diverse metabolites when compared to those from temperate environments [4,5]. This latitudinal hypothesis suggests that chemical defense is mainly driven by predation pressure, and as a result, tropical organisms have evolved to have more effective defenses to deter predators and encroachment, hence having higher chemical diversity [6]. Alternatively, this correlation between source latitude and chemical defense may be attributed in part to a lack of bioprospecting investigations in temperate or polar regions. Studies have shown that organisms from the Antarctic region have comparable levels of bioactivity to those from temperate and perhaps even tropical environments [7]. In addition, Becerro et al. suggested that chemical defenses of tropical and temperate sponges might be equally effective [8].

Tonga has a unique biologically diverse marine environment, influenced by its geographical isolation and the number of major ocean currents in the region. Its marine ecosystem comprises of a number of different and unique species inhabiting the pelagic and coastal areas, including thousands of fish species, marine mammals, turtles, mollusks, crustaceans, urchins, sea cucumbers, and marine algae (seaweeds). A report in 2006 individually identified 202 coral, 150 mollusk, 59 echinoderm, 16 algal, and 54 polychaete worm species in Tongan territorial waters [9]. However, the overall numbers regarding Tongan marine biodiversity is still unknown.

## 4. A Brief History of Marine Natural Products from Tongan Waters

The research group of Professor Philip Crews from the University of California at Santa Cruz (UCSC, Santa Cruz, CA, USA) conducted the first chemical investigations of Tongan marine organisms [10,11,12,13]. This was followed by several natural products studies reported by the Frederick National Laboratory for Cancer Research at Maryland, USA, however, the Coral Reef Research Foundation (CRRF, Koror, Palau) conducted the collections of the organisms [14,15,16]. It should be noted that both organizations were under contract to the United States of America National Cancer Institute (NCI Frederick, MD, USA) to collect marine invertebrates and plants worldwide in search for new naturally occurring anticancer drugs from the ocean. Crews’ research group and the CRRF collection expeditions were made in the 1980s and 1997, respectively, and all the compounds reported from these studies were isolated from specimens collected from the northern Vava‘u group of islands. In addition, all the reported metabolites were obtained simply from marine sponges using bioassay-guided isolation procedures. The apparent lack of chemically driven investigation of Tongan marine fauna presented an opportunity for the research groups of Associate Professors Peter Northcote and Rob Keyzers from Victoria University of Wellington (VUW, Wellington, New Zealand), New Zealand, to shift their research and attention from New Zealand marine organisms to the tropical waters of Tonga. Three collection expeditions were made to Tonga, the first to Tongatapu and ‘Eua in late 2008, followed by the Vava‘u group in late 2009, and a third expedition was to ‘Eua in 2016. These studies employed structure-guided isolation procedures instead of the conventional bioassay-guided isolation methods, which led to the isolation of a wide range of interesting biologically active and new secondary metabolites [17,18,19,20,21,22].

## 5. Tongan Marine Invertebrates

The majority of the organisms collected from Tongan waters for chemical investigation have been marine sponges. Taxonomic identification of sponges has been a difficult task over the years owing to their complex morphological characters and high degree of physical plasticity. Differences in the size and shape of sponges is due to environmental factors, which implies that their shapes are variable among different species and genera, but can also vary between individuals of the same species. Recent techniques have also been used as tools in sponge taxonomic classification such as chemotaxonomic trends and molecular phylogenetics, leading to the reclassification of numerous species [23], several of which are mentioned in this review. However, many of the sponges covered in this review remain unidentified. As much information regarding site description where biota was collected is provided, including depth, but in many cases this data is not available in the published manuscripts.

### 5.1. Sponge-Derived MNPs

#### 5.1.1. Order Poecilosclerida

This undescribed species from the order Poecilosclerida appeared blood-red underwater, and was collected in 1981 from the Vava‘u group [10]. The order Poecilosclerida is the most speciose demosponge order, both in terms of numbers of species and morphology [24]. Poecilosclerid sponges are more common in tropical and subtropical waters, and also occasionally found in temperate and cold, deep waters [24].

Chemistry/Bioactivity: (*S*)-(+)-1-tridecoxy-2,3-propanediol (**1**) displays toxicity to goldfish and was the first novel MNP reported from Tongan sources [10].



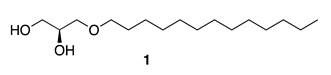



#### 5.1.2. *Diacarnus spinipoculum*

This large, soft, drab sponge was collected from the coral reefs around the island of Nuapapu in the Vava‘u group. This sponge was previously identified as *Prianos* sp. [11] and later revised to *Diacarnus spinipoculum* [25].

Chemistry/Bioactivity: The novel nuapapuin A (**2**) was reported from this specimen [11], together with the known norsesterterpene muqubilin (**3**) [26]. Compound **2** is considered to be the first norditerpene isolated from a marine sponge, and it showed cytotoxic activity against different cancer cell lines such as HeLa human cervix carcinoma (ED_50_ = 16.2 μM), mouse lymphoma L5178Y (ED_50_ = 2.2 μM), and PC12 rat brain tumor (ED_50_ = 18.3 μM) cells [27]. Muqubilin (**3**) was first reported from a sponge of the genus *Prianos* [26], and its absolute configuration was determined in 1985 [28]. A recent article described muqubilin (**3**) as a novel agonist against several key nuclear receptors, proposing **3** as a potential candidate for the treatment of neurological disorders such as Alzheimer’s disease [29].



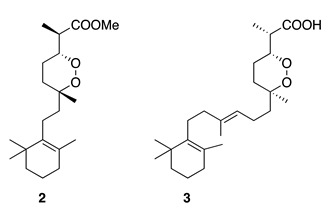



#### 5.1.3. *Hyrtios erectus*

This black sponge is a common inhabitant of the coral reefs around the Vava‘u group and was collected from various locations between 1980 and 1983 [12]. *H. erectus* belongs to the family Thorectidae (order Dictyoceratida) and is widely distributed in the Indo-Pacific [30].

Chemistry/Bioactivity: A study of the anti-inflammatory active extracts from this sponge revealed a novel scalarane norsesterterpenoid, hyrtial (**4**), together with three known sesterterpenes (**5**–**7**) [12]. Hyrtial was subsequently re-isolated with a further five new (**8**–**12**) and two previously described scalaranes (**13**–**14**) [13]. It was during this latter study that **4** was shown to decrease the weight of mouse ear oedema by 43% when inflammation was induced with phorbol myristate acetate (PMA) at a concentration of ca. 50 μg per ear. The authors suggested that compounds **11** and **12** might be artefacts of extraction with methanol. Heteronemin (**5**), originally isolated from the sponge *Heteronema erecta* in 1976 [31], showed potent cytotoxic activity against several human cancer cell lines by disrupting mitochondrial function in a recent published article [32]. Scalaradial (**6**), the first sesterterpenoid with a scalarane skeleton, has also been reported from the sponge *Cacospongia mollior* [33], and exhibits significant anti-inflammatory activity, both in vitro and in vivo, through selective sPLA2 inhibition [34]. Scalarin (**7**) was first reported from the Italian sponge *Cacospongia scalaris* [35], and a recent study by Guzmán and co-workers showed the cytotoxic effect of **7** against several pancreatic cancer cell lines [36].



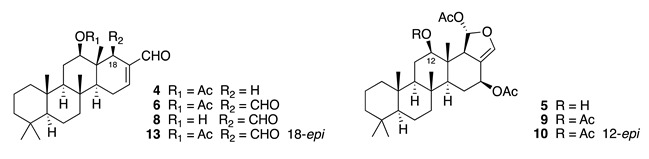



12-Deacetyl-12-*epi*-scalaradial (**8**) has been reported to inhibit the proliferation of several cancer cell lines [37]. A recent isolation of **8** allowed further investigation on its mode of action, which showed it to induce apoptosis in human cervical cancer HeLa cells via extrinsic and MAPK/ERK pathways [38]. Heteronemin acetate (**9**) displayed cytotoxic activity against SKOV3 (IC_50_ = 3.4 μM), SKMEL (IC_50_ = 15.3 μM), BT549 (IC_50_ = 11.2 μM) and Vero (IC_50_ = 8.2 μM) cells [39]. 12-*Epi*-heteronemin (**10**) showed cytotoxicity against human epidermoid carcinoma KB cells (IC_50_ = 5.1 μM) [40]. 12-*Epi*-scalaradial (**13**) has been reported to possess cytotoxic activity against DLD-1 (IC_50_ = 6.1 μM), HCT-116 (IC_50_ = 8.9 μM), T-47D (IC_50_ = 4.7 μM), and K562 (IC_50_ = 4.4 μM) cells [41], while scalarafuran (**14**) exhibited cytotoxic activity against murine lymphoma L1210 (IC_50_ = 6.8 μM) and human epidermoid carcinoma KB (IC_50_ = 9.3 μM) cells [42].



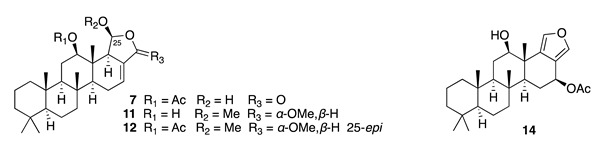



#### 5.1.4. *Pseudoceratina* sp.

This sponge was collected from the Vava‘u group in 1980 and initially identified as *Psammaplysilla* sp. [43]. However, the genus *Psammaplysilla* has since been taxonomically revised to *Pseudoceratina* [44]. This genus belongs to the order Verongiida, which are normally found in tropical to temperate climates.

Chemistry/Bioactivity: Psammaplin A (**15**) is a bromotyrosine metabolite that was reported in 1987, along with the known compound 3-bromo-4-hydroxyphenylacetonitrile (**16**) [43]. Compound **15** is considered to be the first isolated natural product containing both oxime and disulfide moieties. Psammaplin A (**15**) inhibits the activities of several key enzymes in prokaryotic and eukaryotic systems, including those involved in epigenetic control of gene expression, DNA replication, angiogenesis, microbial detoxification, and tumor cell growth [45,46,47,48,49,50,51,52,53,54]. For this reason, psammaplin A (**15**) has become a major research focus for chemists and pharmacologists and has been proposed as a natural prodrug [55].



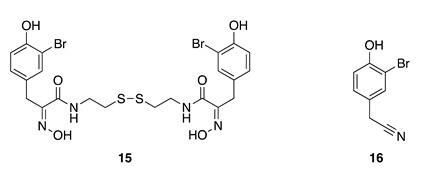



#### 5.1.5. *Halichondria* sp.

Specimens of this undescribed species were collected in Vava‘u by the CRRF under contract with the NCI [14]. The sponge was originally identified as *Pellina* sp., however this genus has since been taxonomically revised to *Halichondria* [44].

Chemistry/Bioactivity: Bioassay-directed fractionations of the organic extract afforded pellynol I (**17**) [14], together with the known compounds pellynols A–D (**18**–**21**) [56] and pellynol F (**22**) [57]. Pellynols A–D (**18**–**21**) were first reported from the marine sponge *Pellina triangulata* [56], while pellynol F (**22**) was first identified from an undescribed *Theonella* sponge [57]. All compounds were shown to display cytotoxic activity against LOX (melanoma) and OVCAR-3 (ovarian) human tumor cell lines. Against LOX cells, pellynols A (**18**), B (**19**), C (**20**), D (**21**), F (**22**), and I (**17**) exhibited antiproliferative effects with IC_50_ values of 0.2–0.8 μM, while presenting IC_50_ values of 2.0–4.5 μM against the OVACAR-3 cell line [14].



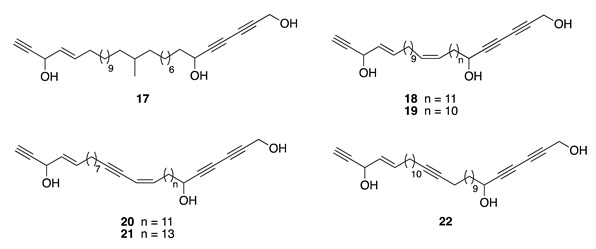



#### 5.1.6. *Jaspis* sp.

In 1997, the CRRF collected a sponge from the genus *Jaspis* in Vava‘u [15]. Sponges belonging to this genus (family Jaspidae) have recently received considerable attention due to them being a rich source of biologically active and structurally novel natural products [58].

Chemistry/Bioactivity: Three isomalabaricane triterpenes, 29-hydroxystelliferin E (**23**), 29-hydroxystelliferin A (**24**), and stelliferin G (**25**), were obtained from the organic extract of the *Jaspis* specimen, along with the known triterpene 3-*epi*-29-hydroxystelliferin E (**26**) [15]. Compounds **24** and **25** were the most growth-inhibitory against the MALME-3M melanoma cell line with IC_50_ values of 0.2 and 0.4 μM, respectively, while both **23** and **26** were approximately 10-fold less potent, with a similar trend observed with MOLT-4 leukemia cells [15].

#### 5.1.7. *Coelocarteria singaporensis*

The sponge was collected by SCUBA at a depth of 7 m in Vava‘u and identified as *Haliclona chrysa* [16], however, this species has since been taxonomically revised to *Coelocarteria singaporensis* [44].

Chemistry/Bioactivity: Isolation of halaminol E (**27**) resulted from utilizing a new automated, high-capacity, and high-throughput procedure developed by the NCI to rapidly isolate and identify biologically active natural products from a pre-fractionated library. Halaminol E (**27**) exhibited low micromolar activity with a GI_50_ value of 6.76 μM against the NCI-60 human tumor cell line panel [16].



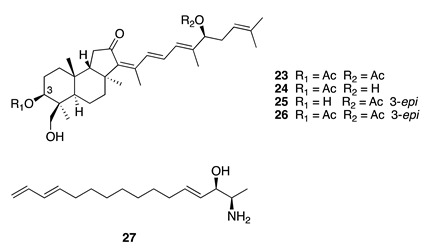



#### 5.1.8. *Plakortis* sp.

This specimen was collected using SCUBA from the ceiling of an underwater cave (depth of 12–15 m) off the coast of ‘Eua Island [18]. This dark purplish brown sponge was identified as a species of *Plakortis* (order Homosclerophorida), which are commonly found in warm waters.

Chemistry/Bioactivity: Spectroscopy-guided chemical analysis of this sponge specimen, afforded seven new metabolites of polyketide origin, lehualides E–K (**28**–**34**), four of which incorporate various sulfur functionalities [18]. The compounds′ structures were elucidated by interpretation of spectroscopic data and spectral comparison with compounds modelling the sulfur-containing functional groups. Lehualides F (**29**) and G (**30**) exhibited growth inhibition of HL-60 cells with IC_50_ values of 6.2 and 5.4 μΜ, respectively, whereas the thioacetate and sulfide metabolites lehualides H (**31**) and I (**32**) displayed weaker inhibition with IC_50_ values of 14.6 and 10.8 μΜ, respectively [18].



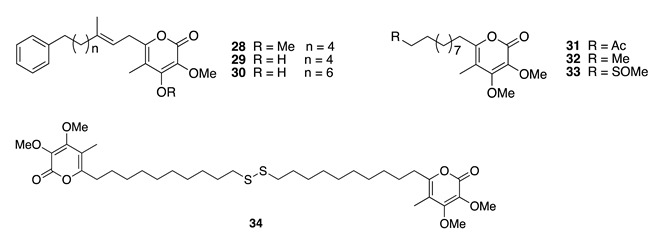



#### 5.1.9. *Strongylodesma tongaensis*

Both the interior and exterior of this massive and soft sponge is black and produced a deep greenish-black mucus on collection (Vava‘u, 2009). To date, *S. tongaensis* has only been reported from three countries (Tonga, Palau and the Federated States of Micronesia) [59]. More concerning, sponges of the genus *Strongylodesma* have previously been taxonomically misidentified as *Batzella*, *Damiria*, *Histodermella*, *Negombata*, *Prianos,* and *Zyzzya*, all reported as sources of pyrroloiminoquinone secondary metabolites [59].

Chemistry/Bioactivity: Initial investigation of this specimen afforded two known pyrroloquinoline derivatives [60], makaluvamine G (**35**) [61] and prianosin B (**36**) [62]. Makaluvamine G (**35**) was originally reported in 1993 by Scheuer et al. from an Indonesian *Zyzzya fuliginosa* (previously identified as *Histodermella* sp.) [61,63], and exhibited general cytotoxicity against a panel of tumor cell lines (IC_50_ 1.2–1.5 μM) while displaying no antifungal or antiviral activities [61]. Prianosin B (**36**), a sulfur-containing alkaloid first obtained from an Okinawan *Prianos melanos*, displayed cytotoxic activity against murine lymphoma cell lines L1210 and L5178Y in vitro with IC_50_ values of 6.0 and 5.4 μM, but only weakly against and the human epidermoid carcinoma KB cell line with an IC_50_ > 15.0 μM [62]. In addition, it was shown to possess moderate cytotoxicity against the HL-60 cell line with an IC_50_ value of 2.2 μM [60].

Re-examination of the same species from the Vava‘u collection yielded two new pyrroloquinoline alkaloids, 6-bromodamirone B (**37**) and makaluvamine W (**38**) [21,64], along with the known compounds makaluvamines A (**39**), C (**40**), E (**41**), F (**42**) [65], I (**43**), K (**44**) [63], makaluvone (**45**) [65], damirone B (**46**) [66], makaluvic acid A (**47**) [67], and tsitsikammamine B (**48**) [68]. Makaluvamine W (**38**) contains an oxazole moiety, which is rare in this large group of natural products, and is the first example of a pyrroloquinoline with nitrogen substitution at C-8. Neither **37** nor **38** were active against the human promyelocytic leukemia cell line HL-60 [21]. This observation was in accordance with the iminoquinone structural requirement necessary for cytotoxicity in the pyrroloiminoquinone alkaloids as reported in the literature [61,67].



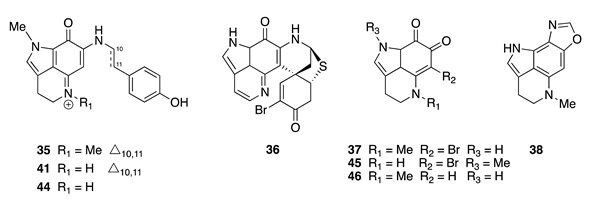



Historically, a series of pyrroloiminoquinone alkaloids were isolated from a Fijian specimen of *Z. fuliginosa* (formerly identified as *Z. massalis*) in 1993, including makaluvamines A (**39**), C (**40**), E (**41**), F (**42**), and makaluvone (**45**) [63,65]. Two years later, makaluvamines I (**43**) and K (**44**) were obtained from a Micronesian specimen of *Z. fuliginosa*, including makaluvamines H, J, L, and M [63]. Damirone B (**46**) was first reported from a Palauan *Z. fuliginosa* (previously identified as *Damiria* sp.) [66], while makaluvic acid A (**47**) was obtained from a Micronesian *Z. fuliginosa* (misspelled in the original manuscript as *Z. fuliginosus*) [67]. Tsitsikammamine B (**48**) was isolated from the related South African Latrunculid sponge, *Tsitsikamma favus* [68]. Compounds **39**, **41**, and **42** were found to exhibit cytotoxicity towards the HCT-116 cell line with IC_50_ values of 1.3, 1.2, and 0.17 μM, respectively, while **45** and **46** were biologically inactive in the same assay [61]. Makaluvamines A (**39**), E (**41**), and K (**44**) exhibited cytotoxicity towards P388 murine leukemia cells (IC_50_ 2.0–2.2 μM) [67], whilst makaluvamine C (**40**) had an IC_50_ value of 2.6 μM towards HL-60 cells [64]. Makaluvic acid A (**47**) displayed no cytotoxicity against murine leukemia P388 cells [67], whereas tsitsikammamine B (**48**) exhibited cytotoxicity against the human colon tumor cell line HCT-116 with an IC_50_ value of 2.4 μM [68].



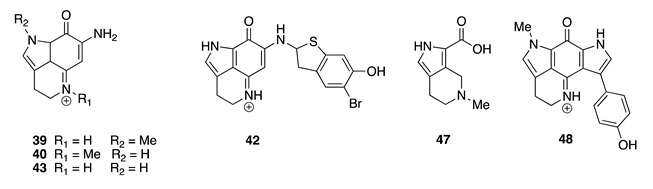



#### 5.1.10. *Cacospongia mycofijiensis*

This species is found throughout the Indo-Pacific with a variable morphology from mushroom to tubular-like shapes, depending on the geographical source location and the surrounding environment [69]. Three collections of this sponge were made by VUW researchers from different geographical locations; two from the island of ‘Eua (2008 and 2016) and one from the Vava‘u group (2009).

Chemistry/Bioactivity: An initial investigation into secondary metabolites of the sponge provided a number of known bioactive and structurally diverse compounds [17] including latrunculin A (**49**) [70], 6,7-epoxylatrunculin A (**50**) [71], dendrolasin (**51**) [72], mycothiazole (**52**) [73], fijianolide A/isolaulimalide (**53**), fijianolide B/laulimalide (**54**) [74,75], neolaulimalide (**55**) [76], and zampanolide (**56**) [77].

Latrunculin A (**49**) is a 16-membered macrolide with an appended 2-thiazolidinone ring, originally reported from the Red Sea sponge *Negombata magnifica* (previously identified as *Latrunculia magnifica*) [70]. Latrunculin A (**49**) is an ichthyotoxic compound and was later found to be an actin polymerization inhibitor, and consequently **49** is the most widely used tool for inhibition of actin polymerization in cell biological studies [78,79]. In 1989, 6,7-epoxylatrunculin A (**50**) was also reported from *N. magnifica* (previously *L. magnifica*) [71]. The biological activity reported for this compound includes cytotoxicity against P388 murine leukemia (IC_50_ = 4.1 μM) and human lung cancer A549 (IC_50_ = 0.5 μM) cells [80]. Dendrolasin (**51**) is a simple furano-sesquiterpene that was originally reported in 1957 from the ant *Dendrolasius fuliginosus* [72], and was found to exhibit extremely weak cytotoxic activity against the human epithelial type 2 cell line HEp-2 [81].



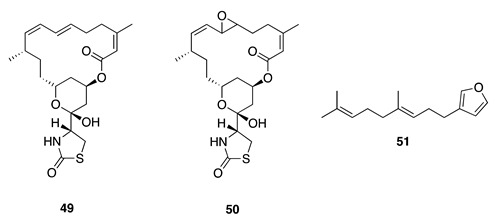



Mycothiazole (**52**) is an unusual heterocyclic polyketide first obtained from a specimen of *C. mycofijiensis* collected from Vanuatu, and was the first disubstituted thiazole reported from a marine sponge [73]. The original chemical structure of **52** was later found to be incorrect, and a corrected structure (the Δ_14,15_ *E* geometry initially proposed was revised to *Z*) was subsequently published in 2006 [82]. Mycothiazole (**52**) displays anthelminthic activity (in vitro) and high toxicity towards mice [72]. Further bioactivity study indicated that it has selective cytotoxicity towards lung cancer cells [82] and also proved it to be a valuable lead mitochondrial complex I inhibitor [83]. A recent article revealed that **52** possesses picomolar potency against PANC-1 (pancreatic), HepG2 (liver), and HCT-116 (colon) cell lines with IC_50_ values of 1.6 × 10^−4^, 2.7 × 10^−4^, and 3.5 × 10^−4^ μM, respectively [84].

From the same Vanuatu collection where they originally reported mycothiazole, Crews and co-workers also identified two cytotoxic macrolides; fijianolides A (**53**) and B (**54**) [74]. These isomers were simultaneously reported as isolaulimalide (**53**) and laulimalide (**54**), respectively, from an Indonesian marine sponge *Hyatella* sp. by Scheuer and colleagues from the University of Hawaii; the latter two names are the more generally accepted by the MNP community [75]. Compound **54** is a potent inhibitor of mammalian cellular proliferation with low nanomolar IC_50_ values, while **53** is considerably less potent with IC_50_ values in the low micromolar range. Both compounds interact with tubulin at a similar but distinct binding site relative to that of paclitaxel [85,86]. The macrolide neolaulimalide (**55**) was first isolated from a Japanese marine sponge *Fasciospongia rimosa*, and its structure was determined from NMR data and by chemical correlation with known congeners [76]. The total synthesis and mode of action was established by Gollner et al. with **55** being demonstrated as a potent microtubule-stabilizing agent [87,88].



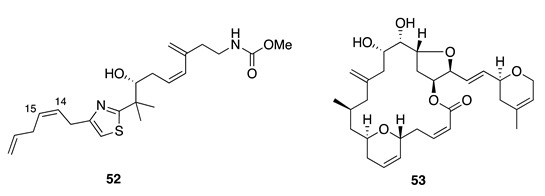



Zampanolide (**56**) was originally reported from the Japanese marine sponge *Fasciospongia rimosa*, and although it exhibited potent cytotoxicity against several cell lines (IC_50_ = 2–10 nM), the mode of action responsible was not determined during the initial investigation [77]. Re-isolation of compound **56** from a Tongan *Cacospongia mycofijiensis* allowed investigation of its mode of action, which showed it to be a novel and potent covalent binding, microtubule-stabilizing compound [17]. Zampanolide (**56**) is cytotoxic at nanomolar concentrations, and arrests cells in the G2/M phase of the cell cycle by irreversible covalent binding to the luminal site of *β*-tubulin, therefore disrupting the function of the microtubule [89]. This places **56** in an important group of anti-cancer compounds that includes the clinically valuable paclitaxel. A continued NMR-guided investigation of the same sponge from a different Tongan collection (`Eua) yielded four new zampanolide analogues, zampanolides B–E (**57**–**60**) [22]. The isolation of these zampanolide analogues gave insight into the structure-activity relationship (SAR) of this family of compounds. Zampanolides B–D (**57**–**59**) exhibited potent antiproliferative activity towards the HL-60 cell line in the low nanomolar range (3–5 nM), and were determined to be potent microtubule-stabilizing agents at levels comparable to zampanolide [22]. Conversely, zampanolide E (**60**), where the key double bond (Δ_8,9_) involved in covalent binding is saturated, was significantly less potent with an IC_50_ value of 306 nM. Surprisingly, zampanolide C (**58**) showed similar activity to the parent compound despite the alteration to the geometry of the Michael-accepting Δ_8,9_ double bond pharmacophore.



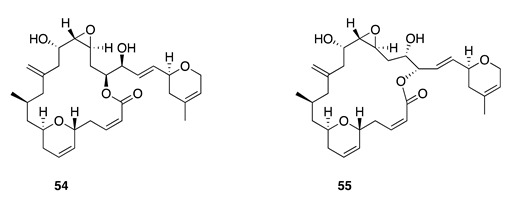



In addition, the re-isolation of dactylolide (**61**) from the same collection established a firm conclusion regarding its controversial absolute configuration, where Tongan-sourced **61** possesses the same absolute configuration as (−)-zampanolide (**56**) and has a levorotatory specific rotation [22], opposite to that for (+)-**61** originally reported from a Vanuatu *Dactylospongia* source [90]. (−)-Dactylolide (**61**) is a microtubule-stabilizing agent and was shown to be slightly more active than **60** despite missing the *N*-acyl hemiaminal side chain, which plays a dramatic role in the activities of these compounds. Alterations to the geometry of the double bonds within the macrocyclic core and side chain of **57**–**59** have no effect on biological activity, suggesting that the side chain and double bond (Δ_8,9_) are both essential for the high potency and microtubule polymerization activity, respectively, of these new analogues [89].



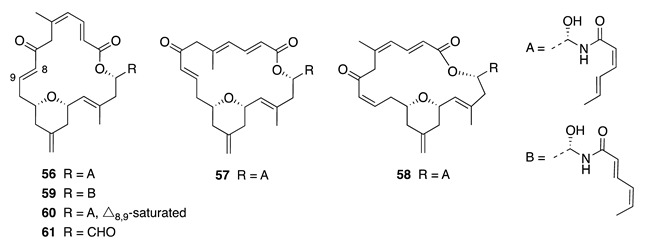



#### 5.1.11. *Fascaplysinopsis* sp.

This massive globular sponge with a shiny red-brown appearance was collected from an underwater cave off the southwestern coast of ‘Eua in 2008. The genus *Fascaplysinopsis* has strong and thick primary fibers and is collagenous throughout the mesophyl of the sponge; specimens are found throughout the Indo-Pacific [91].

Chemistry/Bioactivity: Three known compounds, homofascaplysin A (**62**), isodehydroisoluffariellolide (**63**) [92], and luffariellolide (**64**) [93] were isolated from this specimen [60]. Both **62** and **63** were originally reported in 1991 from the Fijian sponge *Fascaplysinopsis reticulate* [91]. Homofascaplysin A (**62**) was shown to be a potent in vitro inhibitor of chloroquine-susceptible (NF54) and chloroquine-resistant *Plasmodium falciparum* strains with an IC_50_ value of 4.3 × 10^−2^ μM, implying **62** as a promising antimalarial candidate for future drug development [94]. Isodehydroisoluffariellolide (**63**) was found to exhibit cytotoxicity against the HL-60 human promyeloid leukemia cell line with an IC_50_ value of 12.2 μM [60]. Luffariellolide (**64**) was initially obtained from a Palauan *Luffariella* sponge [93]. Shortly thereafter, compound **64** was independently reported from a Fijian *Fascaplysinopsis* sp. and shown to display cytotoxicity against murine lymphoma L1210 and L5178Y (both IC_50_ = 8.5 μM) cell lines [95,96].



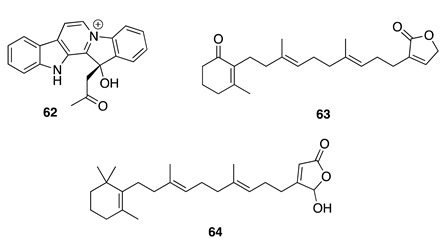



#### 5.1.12. Order Dictyoceratida, Specimen I

An unidentified sponge from order Dictyoceratida was initially collected in 2008 from an unlit cave on ‘Eua, at a depth of ca. 12–15 m while a second collection was made from the Vava‘u group in 2009 [19,97]. The sponge is porous and firm, with both a reticulated surface and oscules. It contains no siliceous spicules and there is little difference in the pigmentation of its pinky-beige exterior and interior [97]. Dictyoceratid sponges do not possess siliceous spicules, which makes their taxonomic identification more difficult.

Chemistry/Bioactivity: Luakuliides A–C (**65**–**67**) were new labdane diterpenes isolated from two samples of the Dictyoceratid sponge collected from ‘Eua, together with the methyl-acetal of luakuliide A (**68**) [19]. These compounds are characterized by a bridging hemi-acetal function on the *B*-ring of the labdane bicycle. Both **65** and **68** displayed very weak inhibition of HL-60 human promyeloid leukemia cells [19].



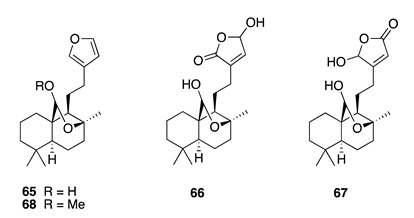



#### 5.1.13. Order Dictyoceratida, Specimen II

This undescribed sponge was collected from an underwater cave at the southwest of ‘Eua that was encrusting with upright fingers with soft slippery texture [60]. It was pale grey on the surface with mid–brownish interior. It contained no siliceous spicules and was tentatively identified as belonging to the order Dictyoceratida.

Chemistry/Bioactivity: Chemical investigation of the methanolic extract of an undescribed Dictyoceratid sponge [60] revealed the known compound thorectolide (**69**) [98]. Compound **69** was first reported from a New Caledonian sponge *Hyrtios* sp. in 1996 and has shown cytotoxic activity against human epidermoid carcinoma KB (IC_50_ = 12.7 μM) [96] and ovarian cancer (1A9) (IC_50_ = 3.7 μM) cell lines [60].



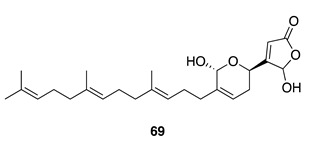



#### 5.1.14. Order Dictyoceratida, Specimen III

This unidentified Dictyoceratid sponge was collected in 2009 from Vava`u in 2009, and it was described as firm but elastic with a dark brown exterior, a green-brown interior and an odor slightly reminiscent of garlic [97].

Chemistry/Bioactivity: Luffariellolide (**64**) [93] was also isolated from this specimen [97], along with the known diterpenoid ambliol B (**70**) [99]. Compound **70** was first reported by Faulkner and Walker from the sponge *Dysidea amblia* in 1981 and contains a *cis*-fused bicyclic ring system [99], which was revised later to a *trans*-fused decalin following X-ray diffraction analysis [100].



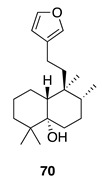



#### 5.1.15. Order Dictyoceratida, Specimen IV

The texture of this sponge, sourced from Vava‘u, was rough, and hard, highly reticulated with a honey-comb shape [101]. The exterior was lilac in color and appeared whitish blue underwater.

Chemistry/Bioactivity: Three known hexahydroxy-9,11-secosteroids, euryspongiols A1, A2, and B1 (**71**–**73**) [102] were isolated [101]. These compounds were originally isolated from a New Caledonian *Euryspongia* sp., along with seven other euryspongiol congeners [102]. Compounds **71** and **72** were found to strongly inhibit the release of histamine from rat mastocysts.



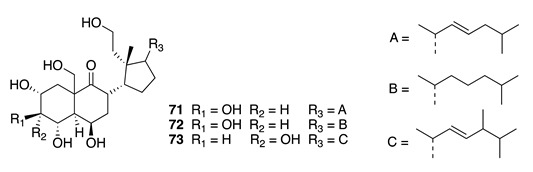



#### 5.1.16. Order Dictyoceratida, Specimen V

This dark brown unidentified sponge was collected from an underwater cave (depth of ca. 20–22 m) on ‘Eua in 2016 [64]. This was an encrusting sponge with large prominent oscula and a pale-yellow interior.

Chemistry/Bioactivity: Examination of this specimen [64] yielded the known compound 6-bromohypaphorine (**74**) [103]. This metabolite was first identified from the sponge *Pachymatisma johnstoni* [103], although it has also been reported from both a marine tunicate [104] and a nudibranch [105]; it is an agonist of the human α7 nicotinic acetylcholine receptor [105].



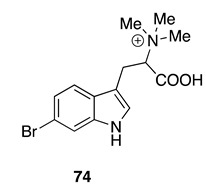



#### 5.1.17. Order Dictyoceratida/Dendroceratida, Specimen I

Less than 1 g of this sponge was collected from the southwestern part of ‘Eua in 2008 [106]. The sponge was small, globular, and intensely blue in color. This sponge was tentatively assigned to either the order Dictyoceratida or Dendroceratida, as these sponges are known producers of sesquiterpenes that lack siliceous spicules, but the size of the specimen precluded further taxonomic identification.

Chemistry/Bioactivity: Guaiazulene (**75**) was obtained from this sponge [106], and although this incredibly blue compound has previously been encountered in gorgonians and terrestrial plants [107], this was its first reported occurrence from a marine sponge.

#### 5.1.18. Order Dictyoceratida/Dendroceratida, Specimen II

This ginger-root like sponge was collected from an underwater cave in the Vava‘u group [64].

Chemistry/Bioactivity: Two known bisabolene-type aromatic sesquiterpenes (**76** and **77**) [108,109] were identified from this specimen [64]. Compound **76** was first reported from the sponge *Halichondria* sp. collected off the Western Australian coast, along with two other aromatic sesquiterpenes [108], while **77** was first reported from the sponge *Didiscus flavus* and displayed cytotoxic activities against P388 murine leukemia and human tumor A549 cell lines, and also inhibited the growth of the fungus *Candida albicans* [109].



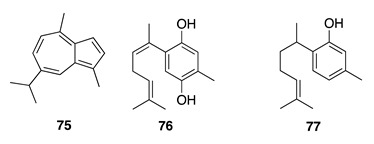



#### 5.1.19. Order Homosclerophorida, Specimen I

This undescribed soft and blood-red colored sponge was collected in 2008 from ‘Eua, and tentatively identified as *Plakortis quasiamphiaster* based on morphological and chemotaxonomic characteristics [106,110].

Chemistry/Bioactivity: Chemical analysis of this sponge yielded the pyrroloacridine compounds plakinidines A (**78**) and B (**79**) [111,112] as the major components [102]. Both compounds were originally isolated from Vanuatu and Fiji collections of the sponge *Plakortis* sp. [111,112], and both exhibited cytotoxicity towards L1210 murine leukemia cells with IC_50_ values of 0.3 and 0.9 μM, respectively [112].



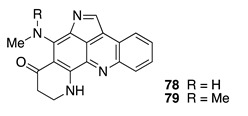



#### 5.1.20. Order Homosclerophorida, Specimen II

The black encrusting sponge was collected at a depth of 12–15 m from an underwater cave on ‘Eua in 2008 [97].

Chemistry/Bioactivity: Two known 5*α*,8*α*-epidioxy sterols (**80** and **81**) [113,114,115,116] were isolated from this undescribed sponge [97]. These metabolites were initially reported by Gunatilaka et al. and have consequently been encountered in a number of different sponge species [113,114,115], the gorgonian *Eunicell cavolini* and the ascidian *Trididemnum inarmatum* [116]. These compounds were evaluated for growth inhibitory effects against MCF-7 human breast cancer cells [116].



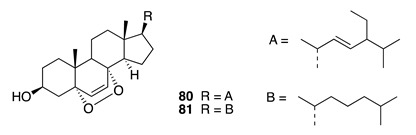



#### 5.1.21. Order Haplosclerida

This sponge was collected from three different locations along the southern coast of ‘Eua in 2008 and 2016. The strong similarity in the morphological and chemotaxonomic profile of these specimens [64,97] and the Fijian sponge *Xestospongia carbonaria* [117,118] led to a tentative identification of the Tongan organisms. However, the sponge *X. carbonaria* was later revised to *Neopetrosia carbonaria* [44].

Chemistry/Bioactivity: Chemical analysis of this specimen from the first collection [97] led to the isolation of the polyketide halenaquinone (**82**) [119,120]. The closely related halenaquinol sulfate (**83**) [119] was identified as the major component of the methanolic extract from the collection in 2016 [64]. The absolute configurations of the two compounds **82** and **83** have previously been established by interpretation of circular dichroism (CD) spectra [119]. Halenaquinone (**82**) is a pentacyclic polyketide first isolated from the marine sponge *Xestospongia exigua* and shown to possess in vitro antibiotic activity against *Staphylococcus aureus* and *Bacillus subtillis* [120]. A recent study, by Takaku et al. described **82** as a novel RAD51 inhibitor that specifically inhibits RAD51-dsDNA binding [121]. Compound **83** inhibited eukaryotic DNA polymerases to varying degrees [122].



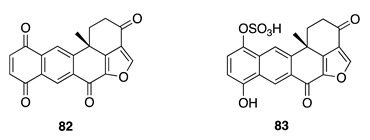



#### 5.1.22. Order Verongiida, Specimen I

The lack of mineral spicules in sponges of the order Verongiida makes their taxonomic identification more difficult. Verongiid sponges show a common and marked oxidative color change at death or upon exposure to air, and they are extremely distinct biochemically, known to be a rich source of bromotyrosine-derived secondary metabolites [24]. An unidentified Verongiid sponge, which oxidized rapidly to dark black at death, was collected from shallow waters using snorkel from a beach in the Vava‘u group in 2009 [60]. Underwater, the sponge was massive, with a yellow-green pigmented surface and yellow interior.

Chemistry/Bioactivity: NMR-guided investigation of this sponge resulted in the isolation of two known bromotyrosine-derived compounds [60], aplysamine-2 (**84**) [123] and aerophobin-1 (**85**) [124]. Compound **84** was originally reported in 1989 from an Australian marine sponge *Aplysina* sp. [123]. Although **84** was reported to be inactive against several Gram-positive and Gram-negative bacteria [123], it revealed weak inhibitory activity against the human tumor cell lines MCF-7 (breast cancer), NCI-H460 (human non-small cell lung cancer) and SF268 (glioblastoma) [125]. Compound **85** was first isolated from the marine sponge *Verongia aerophoba* (later revised to *Aplysina aerophoba* [44]) [124], and exhibited several biological activities including weak inhibition against Factor Xia [126], acetylcholinesterase inhibition (IC_50_ value of 1.3 μM) and antiproliferative activity against MCF-7 cells (IC_50_ value of 0.8 μM) [127].



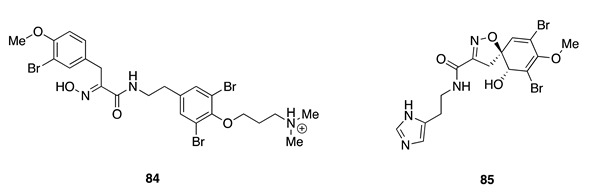



#### 5.1.23. Order Verongiida, Specimen II

This unidentified yellow colored, rubble-like sponge belonging to the order Verongiida was collected from the coastline of ‘Eua in 2008.

Chemistry/Bioactivity: Aerothionin (**86**) [128] is a bromotyrosine-derived compound obtained from this sponge [64]. This compound was first obtained from the sponges *Aplysina aerophoba* and *Verongia thiona* (revised to *Aiolochroia thiona* [44]) [128]. Compound **86** displayed an antifeedant chemical defense role against the predatory fish *Blennius sphinx* [129].



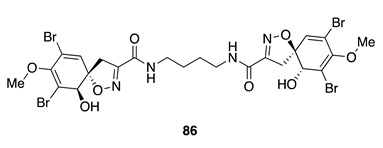



#### 5.1.24. Unidentified Sponge

This undescribed creamy-pink sponge with thin spreading crusts was collected in Vava‘u. Interestingly, upon exposure to air or at death, no change in pigmentation was observed [60]. The sponge was tentatively identified as a Verongiid sponge based on its lack of spicules and its chemical constituents. However, it should be noted that these brominated tyrosine metabolites have been reported from other orders such as Agelasida [130], Dictyoceratida [131], Tetractinellida [132], and Haplosclerida [133] which also may not have spicules. These few counter-examples are a reminder of the complication of using chemotaxonomic markers for sponge classification due to their lack of consistency.

Chemistry/Bioactivity: Fistularin-3 (**87**) [134], aeroplysinin-1 (**88**) [135], LL-PPA216 (**89**) [136], three known bromotyrosine compounds, were obtained from this undescribed specimen [60]. A bioassay-directed isolation procedure first yielded **87** from the marine sponge *Aplysina fistularis* in 1979 [134], however its absolute configuration was only established in a recently published article [137]. It was also shown to display cytotoxic activity against a panel of acute myeloid leukemia (AML) cell lines. Compound **88** was obtained from the methanolic extract of the New Caledonian sponge *Verongia aerophoba* (later revised to *V. cavernicola* [138] and then to *Aplysina cavernicola* [44]) and was the first reported member of this group of alkaloids [135]. Aeroplysinin-1 (**88**) showed antibacterial activity against *Staphylococcus albus*, *Bacillus cereus*, and *B. subtilis* [135], and subsequently shown to have antiproliferative activity in small micromolar doses against several tumor cell lines, including human cervix uterine, Ehrlich ascites tumor (EAT), and HeLa cell lines [139,140,141,142]. LL-PPA216 (**89**) was the first bromo-compound containing two oxazolidone rings, obtained from a *Verongia lacunose* (revised to *Aplysina lacunose* [44]), collected off the coast of Puerto Rico in 1974 [136]. Compound **89** was found to be inactive against several pathogenic bacteria [136] and human cancer cell lines [143].



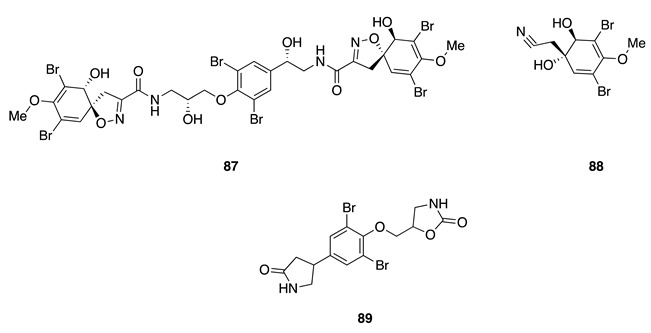



### 5.2. Tunicate (Ascidian) Derived MNPs

#### *Didemnum ternerratum* 

The genus *Didemnum* belongs to the family Didemnidae, which is strongly associated with interesting secondary metabolites [144]. A specimen of this ascidian was collected from an underwater cave (depth of ca. 20–22 m) in ‘Eua in 2016 and is the only tunicate source of Tongan MNPs reported to date [145].

Chemistry/Bioactivity: From this Tongan specimen, six new lamellarin sulfates (**90**–**95**) were reported [145]. NMR and MS experiments were used to elucidate the planar structures of these compounds, while their atropisomeric absolute configurations were determined by comparison of experimental and calculated ECD spectra [145]. All the compounds were tested against the human colon carcinoma cell line HCT-116, where lamellarin D-8-sulfate (**94**) showed some cytotoxicity with an IC_50_ value of 9.7 μM, while the other compounds showed only weak activity [145].



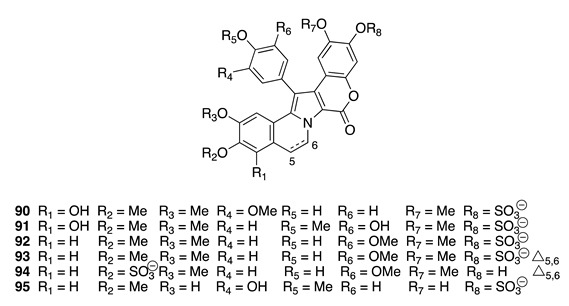



### 5.3. Bryozoan Derived MNPs

#### *Nelliella nelliiformis* 

Bryozoans, also known as sea mats or sea mosses, belong to the phylum Bryozoa with an estimated 8000 described species [146]. Despite the few bryozoan species having been studied for natural products, these organisms have proven to be an excellent source of novel and/or biological active compounds [147]. This brown bryozoan was collected from an underwater cave at a depth of ca. 23 m using SCUBA (‘Eua, 2016) [148].

Chemistry/Bioactivity: A comprehensive chemical examination of this bryozoan afforded two new nucleosides, nelliellosides A (**96**) and B (**97**) [148]. Their planar structures and absolute configurations were determined by interpretation of spectroscopic and chromatographic data and confirmed by total synthesis [148]. Compound **96** was screened against 485 human disease-relevant kinases at a concentration of 10 μM, revealing potent (>80%) and selective inhibition against 13 kinases, while **97** was assessed against seven of these kinases at 10 μM, showing similar levels of kinase inhibition to **96** [148]. Conversely, the two compounds showed no antibacterial or antifungal activities against *Staphylococcus aureus* or *Saccharomyces cerevisiae*, respectively, and neither possessed cytotoxic activity against the HL-60 human cancer cell line [148]. It is noteworthy that this is the first reported chemical investigation of a bryozoan collected from the Kingdom of Tonga, and is the first from this bryozoan family.



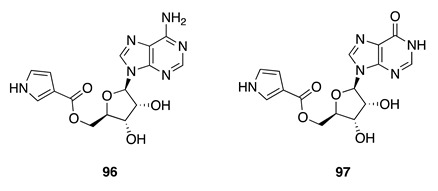



### 5.4. Red Algae-Derived MNPs

#### *Callophycus serratus* 

Red algae of the genus *Callophycus* are widely distributed across Australia, Eastern and Southern Africa, the Philippines, and the South Pacific [149], and are known to be prolific producers of structurally diverse meroditerpenoids [150,151,152,153]. This fern-like red alga was collected from ‘Eua by hand with SCUBA, at a depth of ca. 15–20 m.

Chemistry/Bioactivity: A total of six new halogenated meroditerpenoids (**98**–**103**) were isolated from this red alga [20], along with two known macrolides, bromophycolides A (**104**) [152] and T (**105**) [153]. The relative configurations of the six new compounds across the flexible methylene linker were deduced from detailed analyses of 1D NOE data and ^1^H–^1^H scalar coupling constants. Compounds **100**–**102** incorporate iodine within their structure, which is rare in this family of natural product. Compounds **100**, **104**, and **105** showed moderate cytotoxicity against the HL-60 cell line with IC_50_ values of 5.1, 6.2, and 6.0 μM, respectively [20].



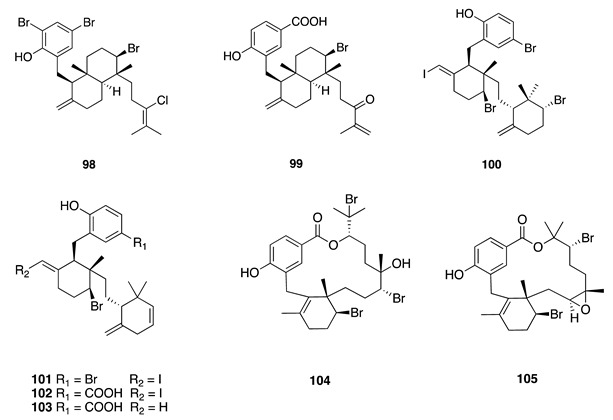



Bromophycolide A (**104**) was first isolated from a Fijian collection of *C. serratus*, along with two other related compounds [152]. The relative configuration for this compound was determined from NMR data while X-ray crystallographic analysis provided the absolute configuration. Compound **104** displayed cytotoxicity against several human tumor cell lines via specific apoptotic cell death. Bromophycolide T (**105**) was also isolated from a Fijian *C. serratus* and was identified by analysis of 1D and 2D NMR spectroscopy and mass spectrometry data [153]. Compound **105** exhibited modest cytotoxicity toward selected human cancer cell lines.

### 5.5. Bacteria-Derived MNPs

#### *Actinomycetospora chlora* Strain SNC-032

This marine-derived bacterium was isolated from a sediment sample collected from a mangrove swamp in Vava‘u [154]. Analysis of its 16S rDNA sequence indicated that the strain was more than 99% identical to *Actinomycetospora chlora.* Marine actinomycetes are prolific producers of biologically active natural products; more than half of the marine microbial secondary metabolites reported in the literature were sourced from actinomycetes [155]. It should be noted that the sample was collected under permits from the Tonga’s Ministry of Agriculture and Food, Forests, and Fisheries, in conjunction with the Northcote and Keyzers group.

Chemistry/Bioactivity: Bioassay-guided chemical investigation revealed three new compounds, thiasporines A–C (**106**–**108**) [154], together with the known compound thiolutin (**109**) [156]. Thiasporine A (**106**) possesses a unique 5-hydroxy-2-phenyl-4*H*-1,3-thiazin-4-one core, the first natural metabolite to possess this motif. Compound **106** displayed cytotoxicity against non-small-cell lung cancer cell line H2122 with an IC_50_ value of 5.4 μM, but no activity against the A549, HCC366, and HCC44 cell lines [154]. However, **107** and **108** were inactive against all the four tested cell lines. Thiolutin (**109**) was originally obtained from the soil bacterium *Streptomyces albus* [156] and shown to inhibit bacterial and eukaryotic transcription in vivo and also used to investigate mRNA stability in several species [157,158,159]. These compounds represent the first secondary metabolites to be reported from marine-sourced bacteria from the South Pacific nation.



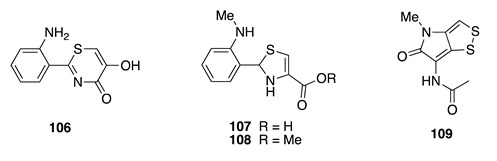



## 6. Conclusions

This review highlights a comprehensive literature survey covering the chemical and biological aspects of secondary metabolites isolated from Tongan marine organisms. A total of 109 compounds were obtained from 24 sponges (81.7%), one ascidian (5.5%), one bryozoan (1.8%), one red alga (7.3%), and one bacterium (3.7%), of which 48 were new MNPs and 61 were previously known metabolites. Known compounds found from Tongan sources were originally reported predominantly from other Pacific nations including Fiji, Vanuatu, New Caledonia, Palau, and Australia. Of the 59 compounds reported from specimens collected in Vava‘u, 40 (67.8%) were known prior to the Tongan investigation, whereas only 21 of 50 (42.0%) of ‘Eua-derived compounds were known. The phylum Porifera (sponges), the main source of Tongan MNPs, was dominated by investigations of the class Demospongiae with the largest number of metabolites (71.6%), followed by the class Homoscleromorpha (10.1%), and no reported compounds from the Hexactinellida or Calcareous sponges. Based on their structural types, there were 42 terpenoids (38.5%), 36 alkaloids (33.0%), 21 polyketides (19.3%), six polyacetylenes (5.5%), two nucleosides (1.8%), one amino alcohol (0.9%), and one glyceride (0.9%). The significant biological activity of Tongan MNPs was dominated by cytotoxicity (70.6%), followed by anti-microbial activity (4.6%).

It should also be noted that more than 60% of the new isolated MNPs were obtained from organisms collected from the island of ‘Eua, despite the fact that the majority of the specimens from these marine natural products studies were collected from the Vava‘u islands group. ‘Eua is the most ancient island in the Kingdom and it is geologically unrelated to the rest of the islands and is believed to be 30 million years older [160]. ‘Eua therefore has a unique marine environment that could harbor organisms that produce interesting and novel chemistry. This became evident during the chemical investigation of *C. mycofijiensis*, with significant geographic variability in the chemistry between the Vava‘u and ‘Eua specimens. The ‘Eua specimens had detectable quantities of zampanolide and possessed new analogues, whereas Vava‘u specimens were significantly less productive in zampanolide with no sign of the new analogues. Thus, bioprospecting ancient island sites within the South Pacific could lead to the discovery of novel compounds with therapeutic potential.

To a large extent the review also perceived the environmental similarity across the Indo-Pacific region on the chemistry of the isolated metabolites, however the change in geographical location induced subtle chemical differences. Chemical analysis of *C. mycofijiensis* specimens collected from Fiji, Vanuatu, and Papua New Guinea yielded similar compounds, however no zampanolide or its analogues were reported, thus revealing the impact of geographic variation upon the chemical composition of these organisms. However, it is also apparent that the Tongan marine organisms have been under-studied from a natural products perspective, considering the large number of marine macro- and microorganisms still remaining to be explored, which may provide viable sources and inspiration for many chemical entities to come.

## Figures and Tables

**Figure 1 molecules-26-04534-f001:**
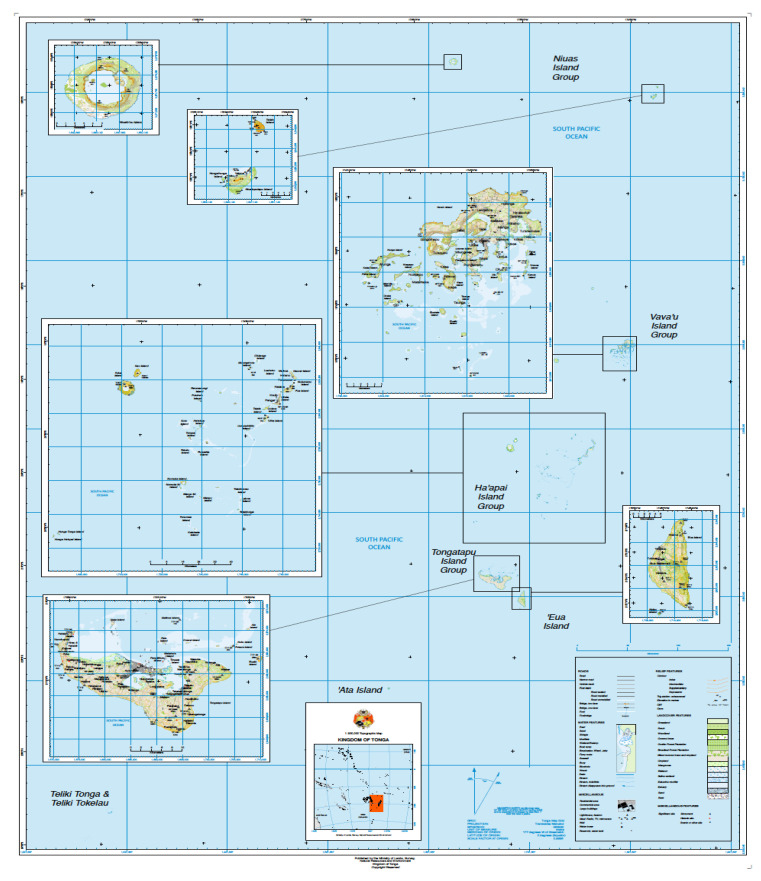
Map of the Kingdom of Tonga. Image courtesy of the Ministry of Lands, Survey and Natural Resources, Tonga.

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
