# Peer review of "Natural Products from Tongan Marine Organisms"

_molecules, 2021, doi:10.3390/molecules26154534_

Round 1

Reviewer 1 Report

The review covers a wide range of naturally occurring compounds in a systematic and comprehensive manner, and provides references to relevant studies where these compounds have been examined for their biological and pharmaceutical properties.  This is a very useful contribution.

In addition to the chemistry and potential application of the compounds described, there is a whole field of study interested in what it is that causes these compounds to be generated, and why they are found in some individuals in high concentrations but not in others.  Some acknowledgment of this would be useful, to advise those researchers interested in that topic that this study does not propose to address these questions, rather it will focus on the chemistry and biological activity.  Perhaps a sentence or two to further clarify the specific intent of the work.

I would be interested to know more detail regarding the locations of the type specimens and the specimens used in the studies, specifically under what light environment (shaded, in a cave, or fully exposed) and the depths.  For those shallow water samples, is the environment turbid and exposed to terrigenous input or could it be considered to be pristine?  These factors may (or may not!) play important roles in the generation of the metabolites examined, which in turn is highly relevant when attempting to determine what may cause the production of these compounds (or the production at different quantities or rates).  The implications of how and why the compounds are generated at all is significant and relevant if harvesting of marine organisms results from the identification of useful compunds.

The paper is timely and well put-together, and I commend the authors for systematically assembling these data.

Author Response

The review covers a wide range of naturally occurring compounds in a systematic and comprehensive manner, and provides references to relevant studies where these compounds have been examined for their biological and pharmaceutical properties.  This is a very useful contribution.

We thank the reviewer for their kind words.

In addition to the chemistry and potential application of the compounds described, there is a whole field of study interested in what it is that causes these compounds to be generated, and why they are found in some individuals in high concentrations but not in others.  Some acknowledgment of this would be useful, to advise those researchers interested in that topic that this study does not propose to address these questions, rather it will focus on the chemistry and biological activity.  Perhaps a sentence or two to further clarify the specific intent of the work.

This is an interesting observation from the reviewer. We have added a line to the end of section 1 that we hope addresses this point (and that of reviewer 3 regarding biosynthesis).

I would be interested to know more detail regarding the locations of the type specimens and the specimens used in the studies, specifically under what light environment (shaded, in a cave, or fully exposed) and the depths.  For those shallow water samples, is the environment turbid and exposed to terrigenous input or could it be considered to be pristine?  These factors may (or may not!) play important roles in the generation of the metabolites examined, which in turn is highly relevant when attempting to determine what may cause the production of these compounds (or the production at different quantities or rates).  The implications of how and why the compounds are generated at all is significant and relevant if harvesting of marine organisms results from the identification of useful compounds.

We have re-examined the primary literature and have included more information regarding the site details for specimen collection, but unfortunately this information is not generally available in most published papers. It is not common to refer to the “state” of the local environment (e.g. pristine vs. contaminated with terrestrial/human inputs) in a manuscript so we cannot directly address this, although as a general comment, Tonga is rather under-developed compared with European/American nations and the environment is generally “pristine”, especially on ‘Eua and in Vava’u where there are no significant industries. A comment has been made to that effect (section 2).

The paper is timely and well put-together, and I commend the authors for systematically assembling these data.

Again, thank you to reviewer 1.

Reviewer 2 Report

This review is related to the natural products (known and new) isolated from marine organisms that were collected within the territorial waters of the Kingdom of Tonga (Indo-Pacific Ocean). The unique biological diversity and geographical isolation of the islands within this archipelago suggests interesting products with different biological activities.

The review is well structured, based on a comprehensive literature survey, well illustrated giving the chemical structures of the isolated compounds and good interpretation of the established biological activities. Although most isolated and characterized compounds are from marine sponges, all other organisms could be sources of biologically active substances.

Taking in account the uniqueness of such regions and the limited access and data about their organisms, I think that this review will be very interesting for the readers of the journal.

Author Response

This review is related to the natural products (known and new) isolated from marine organisms that were collected within the territorial waters of the Kingdom of Tonga (Indo-Pacific Ocean). The unique biological diversity and geographical isolation of the islands within this archipelago suggests interesting products with different biological activities.

The review is well structured, based on a comprehensive literature survey, well-illustrated giving the chemical structures of the isolated compounds and good interpretation of the established biological activities. Although most isolated and characterized compounds are from marine sponges, all other organisms could be sources of biologically active substances.

Taking in account the uniqueness of such regions and the limited access and data about their organisms, I think that this review will be very interesting for the readers of the journal.

We thank reviewer 2 for their kind words. We believe that no corrections/amendments were required by this reviewer.

Reviewer 3 Report

Please  discuss the  similarity and difference between the MNP from Tongan Marine Organisms and from other marine regions.  What compounds were specific to Tongan Marine Organisms?  could you please provide their putative biosynthesis?

Line 144, H. erectus Should be italic

Line 518, change “chemotaxonomic makers” to  “chemotaxonomic markers”.

All the numbers for compounds should be bold.

Author Response

Please discuss the similarity and difference between the MNP from Tongan Marine Organisms and from other marine regions.  What compounds were specific to Tongan Marine Organisms?  could you please provide their putative biosynthesis?

Thank you to this reviewer for their comments. We have added extra text to section 6, lines 631 – 635, to expand upon the prevalence of known compounds, primarily from Vava’u and to a lesser extent from ‘Eua. Additional text has also been provided on lines 657 – 662 with the specific example of Cacospongia mycofijiensis.

Natural product biosynthesis is a large topic of its own right and is beyond the scope of our review; providing biosynthetic reasoning for 109 compounds would expand the review hugely. Additionally, the state of biosynthetic knowledge for invertebrate-derived compounds is largely speculative at this stage, as opposed to the knowledge of microbial natural product biosynthesis where the genes/enzymes responsible are often better characterized. We have therefore left the statement on lines 639 -643 regarding the biosynthetic classes of the compounds but have not expanded the text beyond that.

Line 144, H. erectus Should be italic

Done

Line 518, change “chemotaxonomic makers” to  “chemotaxonomic markers”.

Done

All the numbers for compounds should be bold.

We apologize for this. We have carefully examined our manuscript and have changed all compound numbers to bold typeface. It is possible that we have missed some cases of non-bold typeface, however. There are additional compound numbers to correct, we would be happy to do so.

Reviewer 4 Report

It is a good and interesting review for relevant readers. There are a few comments needed revision before considering acceptance.

  1. The abstract need more revisions, because it is meaningless to introduce the history of  research groups in details. More MNPs details like Line 615-627 should be added in this section.
  2. It is unreasonable to order the main sections (etc. 5. and 6. should be combined together ) at present situation of this manuscript.

Author Response

It is a good and interesting review for relevant readers. There are a few comments needed revision before considering acceptance.

Thanks to this reviewer for their comments.

  1. The abstract needs more revisions, because it is meaningless to introduce the history of research groups in details. More MNPs details like Line 615-627 should be added in this section.

We have altered the text in the abstract, removing the historical information regarding the research groups involved, and have replaced it with some summary information regarding the contents of the review.

  1. It is unreasonable to order the main sections (etc. 5. and 6. should be combined together) at present situation of this manuscript.

Upon reflection we agree with this comment and have combined sections 5 and 6 as requested.